

# Expression pattern of Wif1 and β-catenin during development of anorectum in fetal rats with anorectal malformations

Xiao Bing Tang[1], Huan Li[1], Jin Zhang[1], Wei Lin Wang[1], Zheng Wei Yuan[2] and Yu Zuo Bai[1]

[1] Department of Pediatric Surgery, Shengjing Hospital, China Medical University, Shenyang, Liaoning, China
[2] The Key Laboratory of Health Ministry for Congenital Malformation, Shenyang, Liaoning, China

## ABSTRACT

**Purpose**. This study was performed to investigate the expression pattern of Wnt inhibitory factor 1 (Wif1) and β-catenin during anorectal development in normal and anorectal malformation (ARM) embryos and the possible role of Wif1 and β-catenin in the pathogenesis of ARM.

**Methods**. ARM was induced with ethylenethiourea on the 10th gestational day in rat embryos. Cesarean deliveries were performed to harvest the embryos. The expression pattern of Wif1 and β-catenin protein and mRNA was evaluated in normal rat embryos ($n = 288$) and ARM rat embryos ($n = 306$) from GD13 to GD16 using immunohistochemical staining, Western blot, and real time RT-PCR.

**Results**. Immunohistochemical staining revealed that in normal embryos Wif1 was constantly expressed in the cloaca from GD13 to GD16. On GD13 and GD14, Wif1-immunopositive cells were extensively expressed in the cloaca. On GD15, the expression of Wif1 were mainly detected on the very thin anal membrane. In ARM embryos, the epithelium of the hindgut and urorectal septum demonstrated faint immunostaining for Wif1 from GD14 to GD16. Western blot and real time RT-PCR revealed that Wif1 and β-catenin protein and mRNA expression level was significantly decreased in the ARM groups compared with the normal group on GD14 and GD15 ($p < 0.05$).

**Conclusions**. This study demonstrated that the expression pattern of Wif1 and β-catenin was disrupted in ARM embryos during anorectal morphogenesis, which demonstrated that downregulation of Wif1 and β-catenin at the time of cloacal separation into the primitive rectum and urogenital septum might related to the development of ARM.

Corresponding author
Yu Zuo Bai, baiyz@sj-hospital.org

# INTRODUCTION

Anorectal malformations (ARM) are very common surgical disorders frequently encountered in pediatric surgery practice. The incidence is approximately one in 5,000 live births. There is a wide spectrum of ARM phenotypes, ranging from stenotic anus to cloacal malformation (*Endo et al., 1999*). Surgical operation is the main modality of treatment. Although the level of ARM surgical treatment has improved, there are still different degrees of complications which seriously affect the quality of life (*Peña et al., 1998*; *Peña &*

*Hong, 2000*; *Bai et al., 2000*; *Levitt & Peña, 2005*; *Rintala, 2016*). Up to now, the etiology of ARMs is unknown. Genetic factors are important contributing factors in the pathogenesis of ARMs. Genetic signaling must be precisely regulated in any stage of the hindgut development and its dysregulation contributes to ARMs. *Wif1* is a member of the families of secreted molecules known to inhibit Wnt signalling activity. *Wif1* was first identified as an expressed sequence tag from the human retina, and highly conserved orthologues have been isolated from mouse, Xenopus and zebrafish (*Hsieh et al., 1999*). β-catenin is a critical component of canonical Wnt signaling and is essential for the regulation of cell differentiation and morphogenesis during embryogenesis. The presence of Wif1 leads to β-catenin degradation, thereby turning off Wnt-β-catenin signaling (*Kawano & Kypta, 2003*). Previous study has detected that Wif1 expressed in the midline cloaca endoderm, and dysregulated Wif1 expression caused septation defects. In Wif1$^{lacZ/lacZ}$ mutant mice and cultured urorectum with exogenous Wif1, cloaca septation was defective with undescended urorectal septum (URS) and hypospadias-like phenotypes (*Ng et al., 2014*). Both β-catenin loss- and gain-of-function (LOF and GOF) mutants displayed abnormal clefts in the perineal region and hypoplastic elongation of the URS (*Miyagawa et al., 2014*). These results suggest that Wif1 and β-catenin is required for urorectal development. However, the expression pattern of Wif1 and β-catenin has not been described previously in the embryogenesis of rat ARMs. To provide an insight into the role of Wif1 and β-catenin in anorectal morphogenesis, we have analyzed the expression of Wif1 and β-catenin protein and mRNA in normal and ethylenethiourea (ETU)-induced ARM rat embryos on embryonic stages GD13 to GD16, a critical time in anorectal development.

## MATERIALS AND METHODS

### Animal model and tissue collection

Mature Wistar rats (body weights, 250–300 g) were provided by the Medical Animal Center, Shengjing Hospital of the China Medical University (Shenyang, PR China). Ethical approval was obtained from the China Medical University Animal Ethics (no. 200(7) PS14) prior to the study. Procedures for generating ARMs in fetal rats are described in earlier study (*Bai et al., 2004*). Seventy time-mated pregnant Wistar rats were randomly divided into two groups: ETU-treated group and normal group. In the ETU-treated group, 40 pregnant rats were gavage-fed a single dose of 125 mg/kg of 1% ETU (2-imidazolidinethione; CAS number: 96-45-7; Aldrich Chemical, Penzberg, Germany) on GD10 (GD0 = sperm in vaginal smear after overnight mating). 30 normal rats received corresponding doses of ETU-free saline on GD10. Embryos were harvested by cesarean delivery from GD13 to GD16. One-third of the embryos were fixed in 4% paraformaldehyde for 12 to 24 h depending on their size. Then the embryos from each age group were dehydrated, embedded in paraffin, and sectioned serially sagittally at 4-μm thickness for immunohistochemical staining. The presence of ARMs was determined by light microscope. Then, the embryos were divided into normal and ARM groups. Under magnification, the cloaca/hindgut of the remaining two-thirds of the embryos was dissected and removed from surrounding tissues. The cloaca/hindgut was immediately frozen in liquid nitrogen for Western blot analysis and real-time RT-PCR.

## Immunohistochemical staining

The slides were treated and incubated with primary Anti-Wif1 (1:200 dilution, Rabbit polyclonal, ab186845, UK) and horseradish peroxidase (HRP)-conjugated secondary antibody (Santa Cruz Bio technology, Santa Cruz, CA, USA). Antibody incubations were performed in phosphate-buffered saline (PBS) supplemented with 10% goat serum. Incubation with the secondary antibody was performed for 20 min at room temperature, and signals were visualized by using 3'3Pdiaminobenzidine (DAB; Sigma, UK). Two pathologists independently reviewed the immunohistochemical stained slides and agreed on results by consensus (https://www.protocols.io/view/immunohistochemical-staining-kujcwun).

## Protein preparation and Western blot

Protein preparation was performed as described previously (*Mandhan et al., 2006a*): the cloaca/hindgut per condition were pooled and sonicated in ddH2O containing protease inhibitors. Protein extracts were seperated on SDS-PAGE electrophoresis, and transferred to PVDF membranes, blocked with 5% fat-free milk in Tris-buffered saline (2 h, room temperature). Membrane were incubated in primary antibody against Wif1 (diluted 1:500, Rabbit polyclonal, ab186845; Abcam, Cambridge, UK), β-catenin (diluted 1:1,500, mouse monoclonal, cat#610154; BD Biosciences, San Jose, CA, USA) or anti- β-Actin rabbit monoclonal antibody (1:2,000 dilution; Sigma, St Louis, MO, USA), and incubated with the secondary antibody (diluted 1:3,000, goat anti-rabbit or goat anti-mouse HRP conjugate; Jackson Immunoresearch, West Grove, PA, USA) Membranes were developed by using a chemiluminescent substrate kit (Pierce, Pierce, Rockford, IL, USA) and densitometric values were analyzed by using the ECL Plus detection system (Millipore, Billerica, MA, USA) (https://www.protocols.io/view/western-blot-analysis-kumcwu6).

## RNA isolation and real-time RT-PCR

Total RNA was isolated with the TRIzol reagent (Invitrogen, Carlsbad, CA, USA) according to the manufacturer's protocol. RNA (1 μg) was reverse transcribed by using the Prime Script RT reagent kit (TaKaRa, Shiga, Japan) following the manufacturer's instructions. Quantitative real-time RT-PCR was accomplished with SYBR Premix Ex Tap (TaKaRa, Shiga, Japan) on the 7900HT fast real-time PCR system (Applied Biosystems, Foster City, CA, USA) under the following conditions: 50 °C for 2 min, 95 °C for 10 min, 40 cycles of 95 °C for 15 s, 60 °C for 60 s. A dissociation procedure was performed to generate a melting curve for confirmation of amplification specificity. *GAPDH* was used as the reference gene as reported previously (*Mandhan et al., 2006b*; *Tang et al., 2014a*; *Tang et al., 2014b*). The relative levels of gene expression were represented as $\Delta Ct = Ct$ gene–Ct reference, and the fold-change of gene expression were calculated with the $2^{-\Delta\Delta Ct}$ method. Experiments were repeated in triplicate. The primer sequences spanning the intron-exon junction were as follows:

*Wif1* forward: 5′-AGCCATTCCCGTCAATATCCAC-3′; reverse: 5′-TGCCATGATGCCTTTATCCAG-3′.

*β-catenin* forward: 5′-CGCTTGGCTGAACCGTCACA-3′; reverse:5′-TGGTCCTCGTCATTTAGCAGT-3′.

**Table 1  Distribution of embryos in the various age and treatment groups.**

| Age group | Normal | | | ARMs | | |
|---|---|---|---|---|---|---|
| | IHC | WB | PCR | IHC | WB | PCR |
| GD13 | 25 | 26 | 27 | 30 | 27 | 28 |
| GD14 | 24 | 25 | 26 | 29 | 26 | 27 |
| GD15 | 22 | 25 | 25 | 24 | 25 | 25 |
| GD16 | 20 | 22 | 21 | 21 | 23 | 21 |
| Total | 91 | 98 | 99 | 104 | 101 | 101 |

Notes.
ARMs, anorectal malformations; GD, gestational day; IHC, immunohistochemical staining; WB, Western blot; PCR, real time RT-PCR.

*GAPDH* forward: 5′-GCTGGTCATCAACGGGAAA-3′; reverse:5′ -CGCCAGTAGACTCCACGACAT-3′.

## Statistical analysis

The Statistical Program for Social Sciences, version 13.0 (SPSS, Chicago, IL, USA) was used for statistical analysis. The two-way ANOVA with Post-hoc test was used to compare the Wif1 and β-catenin protein and mRNA levels between the ARM and normal groups. All numerical data were presented a mean ± standard deviation. A value of $p < 0.05$ was considered statistical significance.

# RESULTS

## General observations

In this study, no malformations were observed in the 288 embryos of the normal rats. Among the ETU-treated embryos, all 378 embryos had short or no tail and 19 of embryos died in utero. The incidence of ARMs in ETU-treated embryos was 81.0% (306/378). The embryos for immunohistochemistry staining, Western blot, and real time RT-PCR in each group are shown in Table 1. The type of ARMs was persistent cloaca or rectourethral fistula.

## Immunohistochemical staining
### Normal group

On GD13, the cloaca was divided into urogenital sinus (UGS) ventrally and primitive hindgut dorsally by the L-shaped URS. Wif1-immunopositive cells were extensively expressed on the epithelium and mesenchyme of the cloaca (Figs. 1A, 1B).

On GD14, a potential canal located between the tip of the URS and the cloacal membrane (CM). Wif1-immunopositive cells were detected on the hindgut, URS, urethra and CM (Figs. 2A, 2B).

On GD15, the epithelium on the tip of the URS fused with the dorsal CM, leading to separation of the hindgut and UGS. The anal membrane (AM) was nearly ruptured. Wif1-immunopositive cells were mainly detected on the very thin AM (Figs. 3A, 3B).

On GD16, the AM ruptured and the rectum separated from the UGS completely. The anorectum communicated with the outside. Wif1-immunolabeled cells were observed on the epithelium of the distal anorectum (Figs. 4A, 4B).

## the 13th Gestational Day

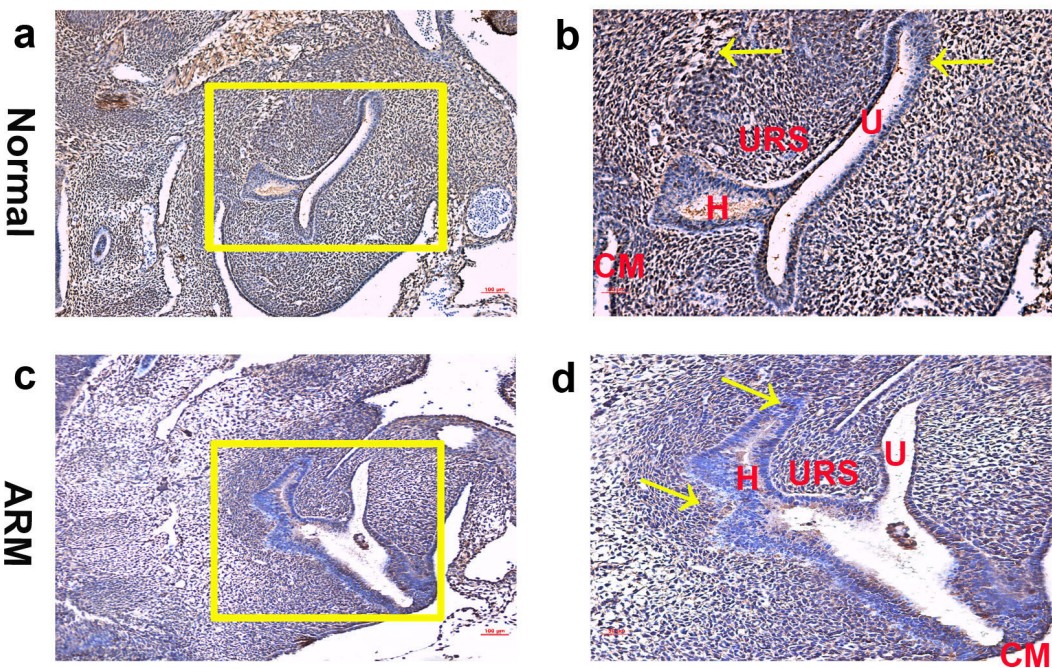

**Figure 1** **Immunohistochemical analysis of Wif1 protein on GD 13.** (A, B) Normal group. On GD13, Wif1-immunopositive cells were extensively expressed on the epithelium and mesenchyme of the cloaca. (C, D) ARM group. On GD13, Wif1-labeled cells were extensively expressed on the epithelium and mesenchyme of the cloaca. (CM cloacal membrane, H hindgut, U urethra, URS urorectal septum). Scale bar = 100 μm in (A, C); = 50 μm in (B, D). Yellow rectangles in (A, C) are shown at higher magnification in (B, D). Original magnification: ×100 (A, C), ×200 (B, D). Yellow arrows in (B) and (D) indicate positive expression of Wif1 protein.

### *ARM group*

On GD13, comparing with normal embryos, the distance between the URS and the CM was long, and the CM was shorter and thicker. Wif1-labeled cells were extensively expressed on the epithelium and mesenchyme of the cloaca (Figs. 1C, 1D).

On GD14, the URS was high in the cloacal cavity, and the distance between URS and CM was relatively long. Wif1 was faintly expressed on the epithelium of the hindgut, URS and the urethra (Figs. 2C, 2D).

On GD15, the distance between the URS and CM shortened, but the URS did not fused with the CM. The fistula between the rectum and urethra was evident, and the hindgut did not separate from UGS. Positive cells were sparsely located on the epithelium of the hindgut, fistula and the urethra (Figs. 3C, 3D).

On GD16, the fistula between the rectum and urethra was existing, and rectal terminus was still not opened to the outside. Wif1 demonstrated low expression on the epithelium of the rectum, fistula and the urethra (Figs. 4C, 4D).

## the 14th Gestational Day

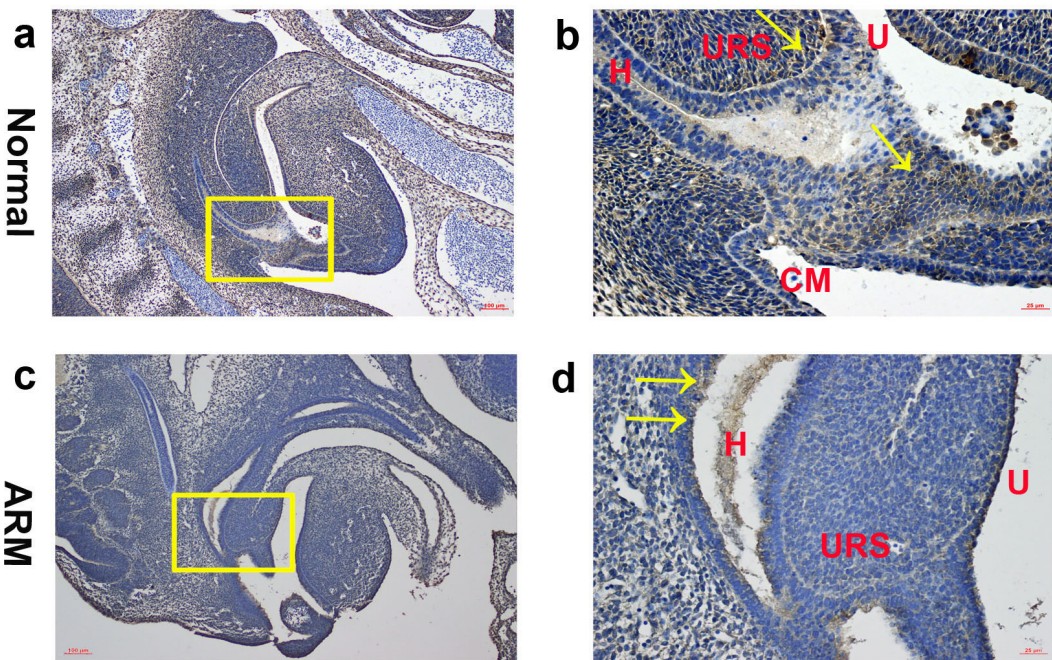

**Figure 2** **Immunohistochemical analysis of Wif1 protein on GD 14.** (A, B) Normal group. On GD14, Wif1-immunopositive cells were detected on the hindgut, urorectal septum, urethra and cloacal membrane. (C, D) ARM group. On GD14, Wif1 was faintly expressed on the epithelium of the hindgut, urorectal septum and the urethra. (CM cloacal membrane, H hindgut, U urethra, URS urorectal septum). Scale bar = 100 µm in A, C; = 25 µm in (B, D). Yellow rectangles in (A, C) are shown at higher magnification in (B, D). Original magnification: ×100 (A, C), ×400 (B, D). Yellow arrows in (B) and (D) indicate positive expression of Wif1 protein.

## Western blot analysis

Western blot specific for Wif1 was performed to quantify protein expression in the anorectal development (From GD13 to GD16: Wif1 protein $1.03 \pm 0.02$ vs $0.94 \pm 0.04$, $1.17 \pm 0.08$ vs $0.99 \pm 0.05$, $1.07 \pm 0.04$ vs $0.91 \pm 0.05$, $0.96 \pm 0.07$ vs $0.90 \pm 0.03$; β-catenin protein $0.98 \pm 0.05$ vs $0.88 \pm 0.02$, $1.14 \pm 0.04$ vs $1.01 \pm 0.05$, $1.09 \pm 0.01$ vs $1.00 \pm 0.05$, $0.87 \pm 0.02$ vs $0.85 \pm 0.01$) (Fig. 5). Wif1 and β-catenin was detected as bands of approximately 41 kDa and 95 kDa respectively among the proteins extracted from normal and ARM tissue. Each protein band was normalized to a corresponding β-Actin band. On GD14 and GD15, the key periods of anus formation, Wif1 expression reached high levels in the normal group but was relatively low in the ARM group($p < 0.05$).

## Real-time RT-PCR

*Wif1* and *β-catenin* mRNA expression was calculated in the normal and ARM groups (From GD13 to GD16: *Wif1* mRNA 1 vs $0.78 \pm 0.24$, $1.29 \pm 0.24$ vs $0.89 \pm 0.07$, $1.26 \pm 0.23$ vs $0.88 \pm 0.03$, $1.12 \pm 0.17$ vs $1.06 \pm 0.13$; *β-catenin* mRNA 1 vs $0.77 \pm 0.12$, $1.13 \pm 0.05$ vs $0.84 \pm 0.14$, $1.05 \pm 0.12$ vs $0.77 \pm 0.12$, $0.77 \pm 0.06$ vs $0.74 \pm 0.09$) (Fig. 6). On GD14 and GD15, *Wif1* and *β-catenin* mRNA expression reached the estimated optimum levels

## the 15th Gestational Day

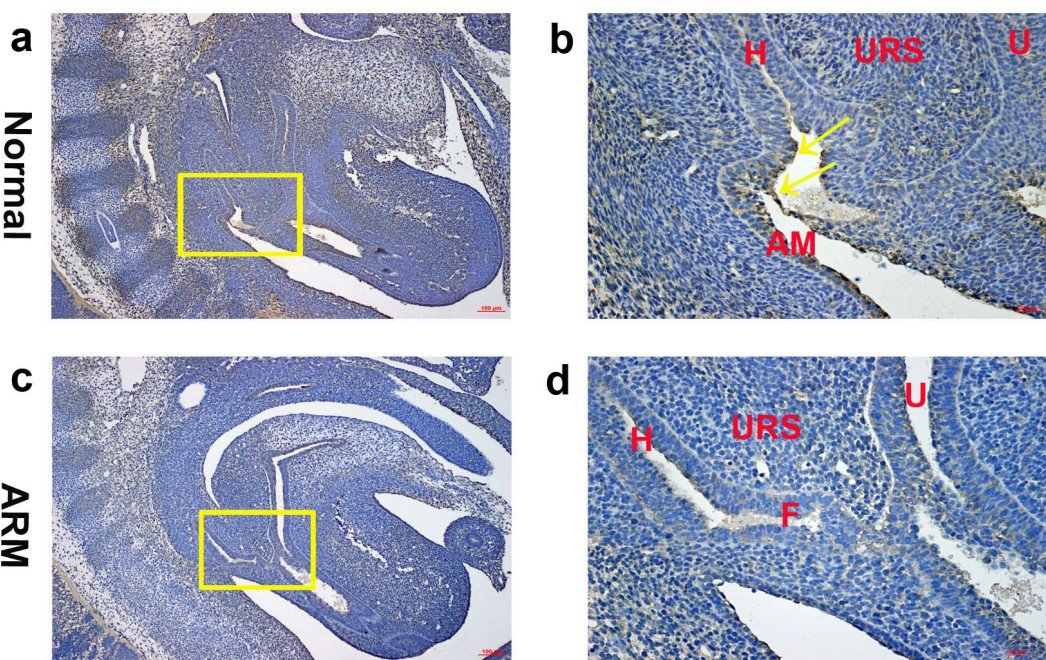

**Figure 3** **Immunohistochemical analysis of Wif1 protein on GD 15.** (A, B) Normal group. On GD15, Wif1-immunopositive cells were mainly detected on the very thin anal membrane. (C, D) ARM group. On GD15, Wif1-positive cells were sparsely located on the epithelium of the hindgut, fistula and the urethra. (AM anal membrane, F fistula, H hindgut, U urethra, URS urorectal septum). Scale bar = 100 μm in (A, C); = 25 μm in (B, D). Yellow rectangles in (A, C) are shown at higher magnification in (B, D). Original magnification: ×100 (A, C), ×400 (B, D). Yellow arrows in (B) indicate positive expression of Wif1 protein.

in the normal group. *Wif1* and *β-catenin* mRNA expression was significantly decreased in the ARM hindgut compared with normal hindgut on GD14 and GD15 ($p < 0.05$).

## DISCUSSION

ETU-induced ARMs in rat embryos has been previously employed to study the morphological changes of ARMs by several groups, including our laboratory (*Qi, Beasley & Frizelle, 2002*; *Bai et al., 2004*; *Mandhan et al., 2006a*; *Mandhan et al., 2006b*; *Zhang et al., 2009*; *Wang et al., 2009*; *Tang et al., 2014a*; *Tang et al., 2014b*; *Zhang et al., 2015*). In this study, expression of Wif1 in the anorectum showed differences in spatial distribution between normal and ARM embryos. Fusion of URS with CM is a traditional theory in the development process of anorectum (*DeVries & Friedland, 1974*; *Qi, Beasley & Frizelle, 2002*; *Bai et al., 2004*). If the URS does not merge with the CM, the anorectum fails to separate from the urethra. A common canal between the rectum and urethra still exists, which leads to the rectourethral fistula or persistent cloaca (types of ARM). On GD15, Wif1-immunopositive cells were selectively detected on the very thin AM (developed from the CM) in normal embryos. In contrast, only sporadic Wif1 immunostaining located

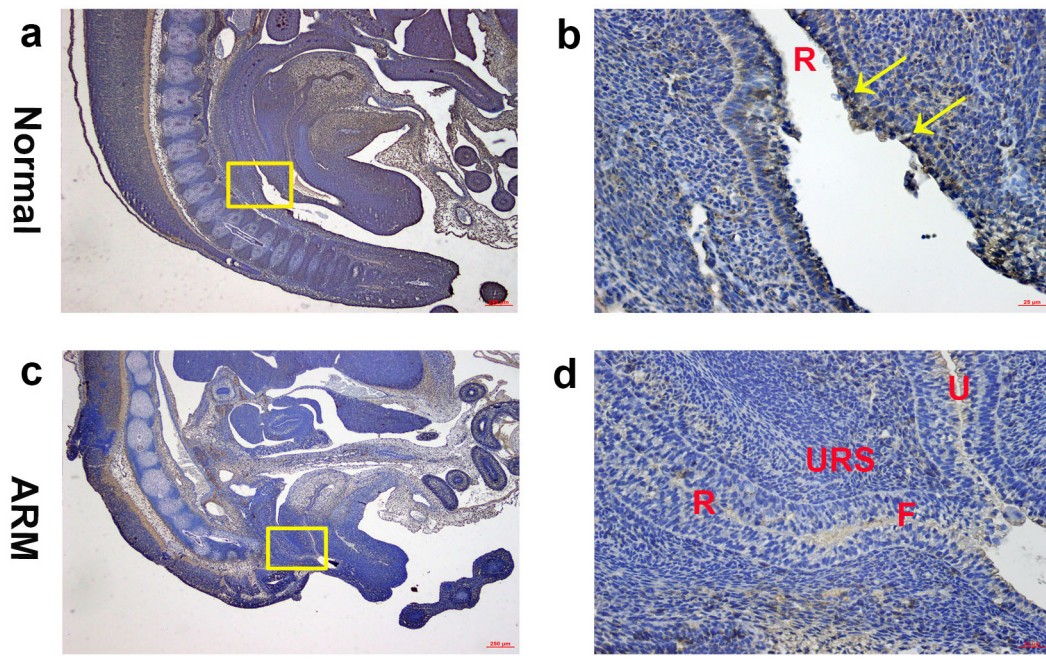

**Figure 4  Immunohistochemical analysis of Wif1 protein on GD 16.** (A, B) Normal group. On GD16, Wif1-immunolabeled cells were observed on the epithelium of the distal anorectum. (C, D) ARM group. On GD16, Wif1 demonstrated low expression on the epithelium of the rectum, fistula and the urethra. (F fistula, R rectum, U urethra, URS urorectal septum). Scale bar = 250 μm in (A, C); = 25 μm in (B, D). Yellow rectangles in (A, C) are shown at higher magnification in (B, D). Original magnification: ×40 (A, C), ×400 (B, D). Yellow arrows in (B) indicate positive expression of Wif1 protein.

on the epithelium of the hindgut, fistula and the urethra in ARM embryos. Therefore, morphogenic events in the anorectum depend on Wif1 signal induction, and Wif1 might be important for the development of the CM during embryogenesis of the anorectum. A recently study indicated WIF-1 suppress the proliferation, invasion and metastasis of the GBC-SD cells and increases the apoptosis of the GBC-SD cells (*Huang et al., 2016*). Lack of Wif1 in CM may decrease the apoptosis of CM and contribute to incompletely separation of the cloaca, thus contributing to the ARMs.

Wif1 and β-catenin expression shows time-dependent changes during anorectal development. Western blot analysis and real time RT-PCR shown that, in the normal embryos, Wif1 and β-catenin expression was at its highest level at the key time-point of anorectal development (GD14 and GD15), suggesting that it may play an role in the development of the anorectum. However, Wif1 and β-catenin expression levels on GD14 and GD15 were significantly lower in the ARM group compared with the normal group, implying that downregulation of Wif1 and β-catenin expression during the critical period of anorectal development may contribute to the ARMs. Additionally, when the anus opened on GD16, the expression of Wif1 and β-catenin protein decreased. This suggest that Wif1 may play an essential role during initial morphogenesis of the anorectum, but its role during subsequent development of the anorectum may be less important.

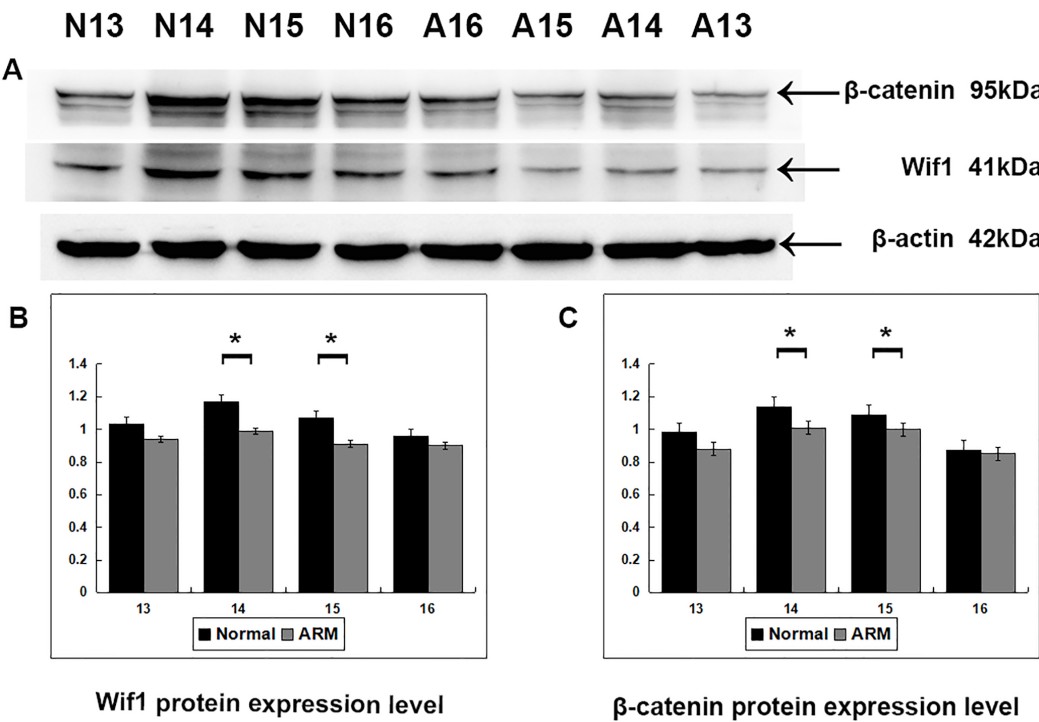

**Figure 5 Western blot analysis of Wif1 and β-catenin protein.** Western blot analysis of Wif1 and β-catenin protein expression levels in normal and ARM developing hindgut tissue samples. Values are presented as means ± SD. (A) Wif1 and β-catenin was detected as bands of approximately 41-kDa and 95 kDa respectively. β-Actin protein is used as an internal control. (B, C) Histogram showing the trends of Wif1 and β-catenin expression at each time-point. A peak can be noted on GD14 and GD15, by Tow Way ANOVA and *post hoc* test.

Wif1 is one member of Wnt antagonists, which bind to Wnt directly and inhibit the link with their receptors, and as a result, the accumulation of β-catenin is reduced and canonal and noncanonal pathway are inhibited (*Malinauskas et al., 2011*). *Ng et al. (2014)* detected that Wif1 expressed in the midline cloaca endoderm, and dysregulated Wif1 expression caused septation defects. β-catenin LOF and GOF mutants both displayed abnormal separation of the cloaca and maldevelopment of the URS (*Miyagawa et al., 2014*) β-catenin is a critical component of canonical Wnt signaling and is essential for the regulation of cell differentiation and morphogenesis during embryogenesis. Dysregulation of Wif1-β-catenin signaling may influence proliferation and apoptosis of CM and URS, and lead to maldevelopment of the anorectum, even contributing to ARM.

ETU is known to disturb the expression of the shh signaling pathway during the development of the hindgut (*Mandhan et al., 2006a*; *Mandhan et al., 2006b*). RC-L Ng had demonstrated that dysregulation of Shh-Wif1-Wnt-β-catenin signaling contributes to ARMs using transgenic mice (*Ng et al., 2014*). Wif1 and β-catenin levels were reduced in ETU exposed embryos during hindgut development. β-catenin signaling is required for caudal neural tube closure and elongation, acting through the transcriptional regulation of Cdx2 (*Zhao et al., 2014*). β-catenin GOF mutants ectopically induces Bmp4 and Bmp7

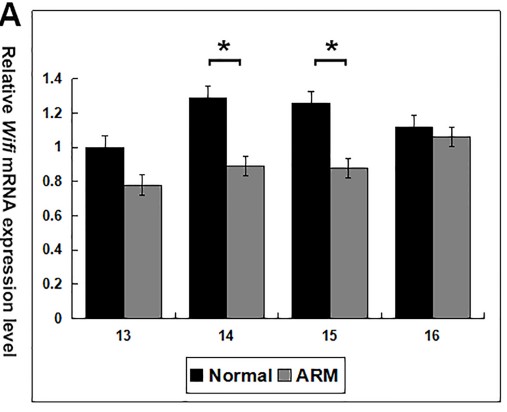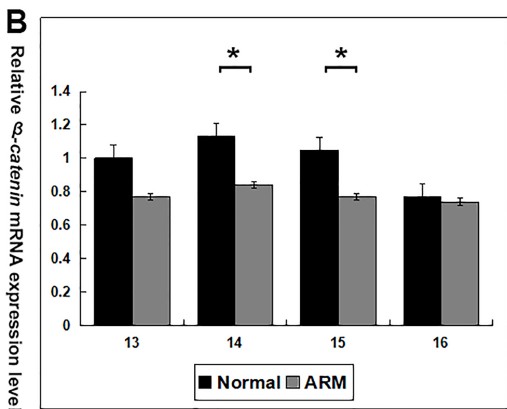

**Figure 6    Real-time RT-PCR analysis of *Wif1* and *β-catenin* mRNA expression levels in normal and ARM-developing hindgut tissue samples.** On GD14 and GD15, the key period of anus formation, *Wif1* and *β-catenin* expression reaches the estimated optimum levels in the normal group, whereas in the ARM group, *Wif1* and *β-catenin* mRNA is minimally expressed. Values are presented as means ± SD. ∗ Significant difference from corresponding controls, by Tow Way ANOVA and *post hoc* test.

expression in the epithelia of the URS, and the ARM phenotypes in the β-catenin GOF mutants could be restored by additionally introducing the Bmpr1A gene mutation (*Miyagawa et al., 2014*). Previously studies showed that ETU disturbed the expression of Cdx2, Bmp2, Bmpr1a, Bmp4, Bmp7 in anorectum of rat embryos (*Mandhan et al., 2006b*; *Tang et al., 2014b*; *Tang et al., 2016*; *Zhang et al., 2015*). All the above evidences indicate that ETU might disturb the expression of Cdx2 and Bmp through Wif1-β-catenin signal pathway and lead to ARM in rat embryos. Cdx2 and Bmp may function downstream of Wif1-β-catenin signal pathway during anorectal morphogenesis.

## CONCLUSIONS

In summary, the expression pattern of Wif1 and β-catenin was impaired during development of anorectum in fetal rats with ETU-induced ARMs. This indicates that Wif1 and β-catenin might play an important role in morphogenesis of the anorectum. Decreased Wif1 and β-catenin expression might be related to the development of ARMs. Further studies are needed to define the specific roles of Wif1 and β-catenin during anorectal development, and thus improve our understanding of the pathogenesis of ARMs.

### Funding

This study was supported by the National Natural Science Foundation of China (grant numbers 81600402, 81470788), the Project of Key Laboratory of the Education Department of Liaoning Province (grant number LS201601) and the Outstanding Scientific Fund of Shengjing Hospital (grant number 201502). The funders had no role in study design, data collection and analysis, decision to publish, or preparation of the manuscript.

## Grant Disclosures

The following grant information was disclosed by the authors:
National Natural Science Foundation of China: 81600402, 81470788.
Project of Key Laboratory: LS201601.
Outstanding Scientific Fund of Shengjing Hospital: 201502.

## Competing Interests

The authors declare there are no competing interests.

## Author Contributions

- Xiao Bing Tang performed the experiments, prepared figures and/or tables, authored or reviewed drafts of the paper, approved the final draft.
- Huan Li performed the experiments, prepared figures and/or tables, authored or reviewed drafts of the paper, approved the final draft.
- Jin Zhang performed the experiments, authored or reviewed drafts of the paper, approved the final draft.
- Wei Lin Wang and Zheng Wei Yuan analyzed the data, authored or reviewed drafts of the paper, approved the final draft.
- Yu Zuo Bai conceived and designed the experiments, contributed reagents/materials/analysis tools, authored or reviewed drafts of the paper, approved the final draft.

## Animal Ethics

The following information was supplied relating to ethical approvals (i.e., approving body and any reference numbers):

Ethical approval was obtained from the China Medical University Animal Ethics Committee (no. 200(7) PS14) prior to the study.

## Data Availability

The raw data is provided in the Supplemental Files.

## Supplemental Information

Supplemental information for this article can be found online at http://dx.doi.org/10.7717/peerj.4445#supplemental-information.

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
