# Peer review of "Expression pattern of Wif1 and β-catenin during development of anorectum in fetal rats with anorectal malformations"

_PeerJ, doi:10.7717/peerj.4445_

## Round 0.1 · original submission · Major Revisions

Dear Authors

Please have a look at the reviewer's thorough comments and correct what is asked for, including performing a few of the extra experiments that are suggested by the reviewers.

Reviewer 1 ·

Basic reporting

Tang and co-workers have submitted a manuscript dealing with the expression and potential function of Wif1 in the anorectal region of rats. Wif1 is a Wnt-signaling regulator (suppressor) and known to play a role in mammalian development. Its role in anorectal development is of general interest, particularly if it plays a role in disease development, such as ARM (anorectal malformations), a relatively common disorder in human newborns.

The manuscript is presented in a relatively clear and logical manner and deals with a topic of relevance. Regarding the content, it is relatively minimalistic, showing the expression pattern of one factor (Wif1) in normal versus ARM rats at critical developmental stages. The authors show that Wif1 is dysregulated in ARM rats and from there, conclude that Wif1 could be involved in ARM pathogenesis. However, no effort is made towards deciphering how Wif1 might possibly be functionally involved, leaving the conclusion speculative without any empirical data to support a causal rather than effectual relationship between Wif1 downregulation and ARM phenotype.

A previous study by another group has already reported on a possible involvement of Wif1 in ARM pathogenesis, using mouse models to not only show dysregulated Wif1 expression, but downstream consequences where the Wnt/b-Catenin signaling pathways is disturbed. Although the present study is using the rat as a model, there is little to suggest that Wif1 function in anorectal development is in any way different between the two closely related rodent species. Thus, this manuscript only corroborates the finding that Wif1 is dysregulated in ARM, but does not corroborate additional functional findings, nor provide any additional knowledge about Wif1 function.

Experimental design

see 'general comments'

Validity of the findings

see 'general comments'

Additional comments

Specific Comments:
1) The language and structure would benefit from further editing.

2) Additional experiments should be performed to support the view that WIF1 is functionally involved in urorectal development or ARM pathogenesis, not least since expression pattern and function of Wif1 has already been shown in comparative tissues in mice. Other factors should be analysed, such as Wnt/b-Cat signaling (as a minimum).

3) Introduction should introduce properly what is known about Wif1 in anorectal development, and ARM, referring particularly to the paper by Ng et al (2014). Presently, the text reads that “…dysregulated Wif1 expression caused septation defects (Ng et al 2014). These results suggest that Wif1 is required for urorectal development. However, the expression pattern of Wif1 has not been described previously in the embryogenesis of rat ARMs.” This is incomplete and misleading information.

4) The discussion generally lacks references, makes unspecific statements and contains a lot of repetition

5) Please use standard nomenclature for gene and protein names

6) Materials & Methods: the group of rats that was ETU-free saline is referred to as “control” whereas it is referred to as “normal group” in the remainder of the paper. Use same denomination throughout.

7) For antibodies, the catalogue numbers should be listed, as there are more than one available for WIF1. Figues also appears to be counterstained. This should be described in the protocol (as well as explained in Fig legends). Also list CAS number for 2-imidazolidinethione.

8) Refer to Table 1 in Mat & Meth section where number of embryos for different experiments are reported.

9) In line 75 it is stated that “One third of the embryos were fixed” while line 79 states “cloaca/hindgut of other specimens was dissected” does this then refer to the remaining two thirds? Please clarify.

10) Gapdh was used as reference gene in qPCR experiments. Data confirming Gapdh is stably expressed under the experimental conditions should be provided, or a reference to other work having shown so. Further on qPCR results. Fig 6, it is unclear what samples the significant asterix denotations refer to. As it is depicted, one would assume controls at GD14 and GD15 are significantly upregulated in relation to control GD13, whereas I believe the authots mean to show difference between control and ARM samples at single time points. If so, the atserisks should be over ARM bars, or better still, with a line connecting the two samples the asterisk refers to.

11) Western blot data in Fig 5 is not overly convincing, but ok when taking observations from IHC into account, But as commented upon under qPCR fig (above), the way the asterisks are shown, it is unclear (or wrong) what samples the statistical difference refer to. This must be corrected. In Figs 5-6, also include statistical method used and what significance/confidence level the asterisk refers to. Also, the supplementary blots only show the cropped images. The full blots should be provided.

12) All Figures: legends should contain more information, such topic/main finding, all marker stains, and n-numbers, in addition to what is already included.

13) Lines 156, 160-161 & Figure 6: What is meant by “Optimal/optimum levels” or “Minimally expressed”?

14) Line 155 & 161: GD13 should be corrected to GD14

15) Tables 2 and 3 are redundant.

Reviewer 2 ·

Basic reporting

The authors used ARM animal model to study the potential relationship between Wif1 expression and ARM. They found both the protein and RNA level of Wif1 were reduced in the ARM rats and concluded that Wif1 may play a role in the development of ARM. Overall, the text is fluent and this work is interesting in regard to the expression pattern of Wif1 in ARM rats. However, I still have some concerns.
1. The authors used t test to analyze the data. However, two-way ANOVA with Post-hoc test is more appropriate. The authors should re-analyze the data.
2. The authors tested the protein and RNA level of Wif1 in the cloaca/hindgut. They should also test the Wif1 level in other tissues. Since the authors used ETU to treat the pregnant rats, it is important to know whether the reaction of Wif1 is universal in the embryos or specific in certain tissues.
3. Since the expression level of Wif1 was changed in the ARM rats, the authors should test at least one downstream gene of this signaling pathway to further demonstrate the disruption of the signaling.
4. I have a hard time to evaluate Figure 1,2,3,4. It seems to me that there is no reduction of Wif1 in ARM rats. Could the authors use arrows to indicate the differences between the normal and ARM images?
5. As the authors mentioned in the text, previous study has already showed that Wif1 may play a role in the development of ARM (Ng et al. 2014). The authors should compare their studies and discuss more carefully.
6. Line 209-213: I think it is inappropriate to make such a conclusion in the text since the authors only find the expression change without any further investigation.
7. The images in Figure 1, 2, 3a, 3b are different from the raw data that the authors uploaded. They need to clarify this.
8. It is hard to evaluate the raw data of westernblot. The authors should upload the raw data of full-gel images with the protein ladders and proper labeling.

Experimental design

no comments

Validity of the findings

no comments

Additional comments

no comments

---

## Round 0.2 · accepted · Accept

I have checked your resubmission. It is great that you have now included catenin expression.